# A rapid and accurate protocol for quantifying living *Acanthamoeba castellanii* cysts

Kazushi Matsubara,[1] Ryohei Hirose,[1,2] Norihide Hasegawa,[1] Takumi Minamiyama,[1,3] Taku Kano,[1,2] Akinobu Sai,[1,2] Minoru Yamada,[1] Takaaki Nakaya[1]

**ABSTRACT** The accurate quantification of living *Acanthamoeba castellanii* (AC) cysts after various interventions is essential for evaluating the effectiveness of AC cyst disinfection and their environmental stability. The current protocol for measuring living AC cysts requires 7 days to complete, posing a substantial challenge. In this study, we aimed to develop a rapid protocol that markedly reduces the time required to accurately quantify living AC cysts. Clinical and standard strain AC cysts were inoculated into 96-well plates and incubated in peptone-yeast extract-glucose (PYG) medium with varying concentrations of fetal bovine serum (FBS) and $CO_2$ levels. Excystation was monitored under an inverted microscope, and the living AC cyst counts (LACC) were calculated. We found that the excystation of AC cysts was enhanced by increasing the FBS concentration in the PYG medium and increasing the $CO_2$ concentration during culture, which collectively significantly reduced the duration required for quantification. Specifically, by supplementing PYG medium with 5% or more FBS and incubating at 2.5% $CO_2$ or higher, measurements could be completed within only 3 days for standard and clinical AC strain cysts. Notably, this rapid protocol maintained the same accuracy as the conventional 7 day protocol and provided high accuracy in the LACC assay for samples exposed to disinfectants. In summary, the rapid protocol we developed can reduce the time required for measurement by more than half compared to current protocols and will contribute to substantially expediting the evaluation of the effectiveness of disinfectants and other drugs against AC cysts.

**IMPORTANCE** *Acanthamoeba castellanii* (AC) is a pathogenic microbe that causes refractory *Acanthamoeba* keratitis (AK). AC cysts are highly durable and resistant to various disinfectants, making effective disinfection, which is essential for mitigating AK, difficult. Currently, it takes 7 days to determine disinfection effectiveness against AC cysts because it involves measuring viable AC cyst counts after disinfection. However, the novel, accurate method for quantifying living AC cysts developed in this study can be completed within 3 days, reducing the time required for measurement by more than half compared to conventional methods, and will markedly streamline the evaluation of disinfectant effectiveness against AC cysts. Moreover, this quantitative method can potentially be applied to shorten the time required for quantifying living cysts of other amoebas.

**KEYWORDS** *Acanthamoeba castellanii*, excyst, carbon dioxide, fetal bovine serum, quantification

*A*canthamoeba is a protist widely distributed in soil, freshwater, and other environments and is a pathogen that causes refractory keratitis and granulomatous amebic encephalitis (1, 2). The life cycle of *Acanthamoeba* includes two forms: trophozoites, which grow actively, and cysts, which do not grow and exhibit minimal metabolic

Address correspondence to Ryohei Hirose, ryo-hiro@koto.kpu-m.ac.jp.

Kazushi Matsubara and Ryohei Hirose contributed equally to this article. Author order was determined based on the amount of data collected and analyzed by each author.

The authors declare no conflict of interest.

See the funding table on p. 13.

activity. When exposed to harsh environments, trophozoites differentiate into cysts and develop resistance to several antibacterial/antifungal drugs and disinfectants, as well as harsh environments (1–6).

*Acanthamoeba castellanii* (AC) is an opportunistic pathogen and primarily causes refractory *Acanthamoeba* keratitis (AK). AK is typically transmitted through contact lenses contaminated with AC; therefore, to prevent AK, it is vital to effectively disinfect (kill) AC trophozoites and cysts in contact lenses and storage containers (7–10). However, as AC cysts are highly resistant to various disinfectants, it is difficult to achieve effective disinfection (5–7), and this poses a challenge that is yet to be resolved and warrants further research (11).

The accurate quantification of living AC cysts after various interventions is essential for evaluating the effectiveness of disinfection against AC cysts as well as their environmental stability. Although the viability of AC trophozoites can be rapidly determined using conventional cell viability assays based on measuring metabolic activity or ATP content (12, 13), these assays cannot be applied to AC cysts. Recently, Veugen and Wolffs developed a sensitive viability PCR assay using a photoreactive dye for rapid detection of living AC cysts (14). This assay can detect living AC cysts in an extremely short time; however, there are still barriers to achieving absolute quantification based on this method. Ultimately, long-term culture and visual confirmation of cyst excystation are still important for accurate absolute quantification of living cysts that have the potential to excyst and proliferate (i.e., those demonstrating pathogenicity) (15, 16). The quantification of living cysts is typically performed using two methods: quantification on agar medium (plaque assay) and quantification in liquid medium. We focus on the latter method because the two methods share similar concepts, accuracy, and measurement times. In this method, serially diluted samples are cultured on a 96-well plate for 7 days, and the presence of trophozoites from excysting cysts is observed under a microscope (Fig. 1). The number of living cysts is calculated from the wells containing trophozoites using the Reed-Muench or Spearman-Karber methods (15, 17–21). This quantification method is similar to the calculation methods used for 50% lethal dose and 50% tissue culture infectious dose ($TCID_{50}$). For example, the $TCID_{50}$ assay, widely used for virus quantification, is suitable for quantifying low-titer virus samples and can accurately determine if a virus has been completely inactivated (22, 23). Therefore, the $TCID_{50}$ assay is also applicable for evaluating disinfection effectiveness and environmental stability (24, 25). Similarly, the abovementioned conventional method for quantifying living cysts is considered accurate and suitable for evaluating AC cyst disinfection and environmental stability. However, its accuracy has not been thoroughly evaluated. Therefore, this study aimed to determine the accuracy of this living cyst quantification method.

The current protocol for quantifying living AC cysts requires 7 days to complete, posing a substantial challenge. To shorten this time, it is necessary to accelerate the excystation of AC cysts. Siddiqui, Lakhundi, and Khan investigated the influence of environmental and physiological conditions on AC cyst excystation, suggesting that it could be accelerated by optimizing the medium pH and incubation temperature and in the presence of fetal bovine serum (FBS) and $CO_2$ (26). Building on their findings, in this study, we quantified living AC cysts by culturing cysts under varying FBS and $CO_2$ concentrations and evaluated the time required for measurement. Ultimately, we aimed to develop an accurate method for quantifying living AC cysts that required significantly less time than the conventional measurement method.

## RESULTS

### Excystation and growth rate under various culture conditions

The excystation and proliferation of the AC standard strain (ATCC 50492) cysts under various culture conditions were observed over time. On day 2 of incubation, although minimal excystation was observed when cysts were cultured with 0.04% $CO_2$ (i.e., 100% air), excystation increased slightly when cysts were cultured in peptone-yeast extract-glucose (PYG) medium supplemented with 5% FBS. However, excystation was

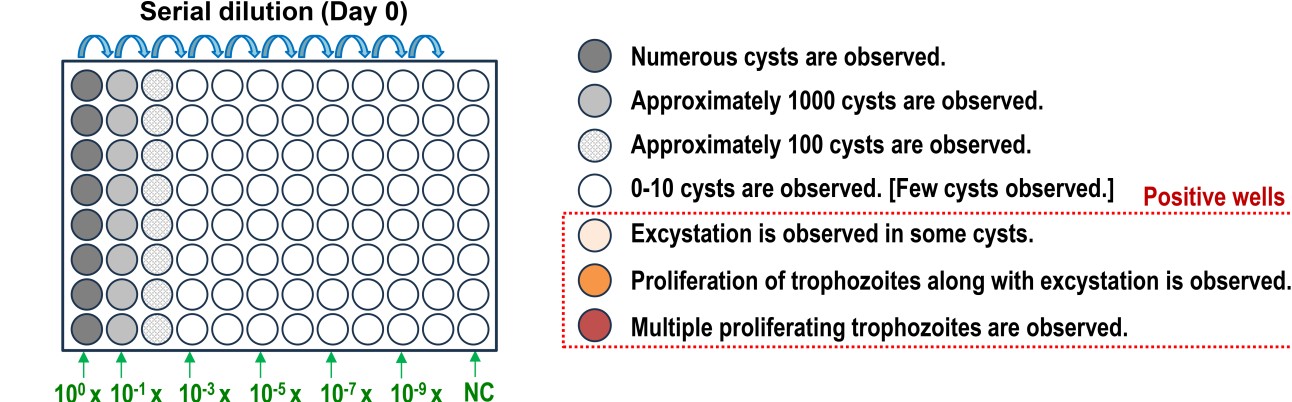

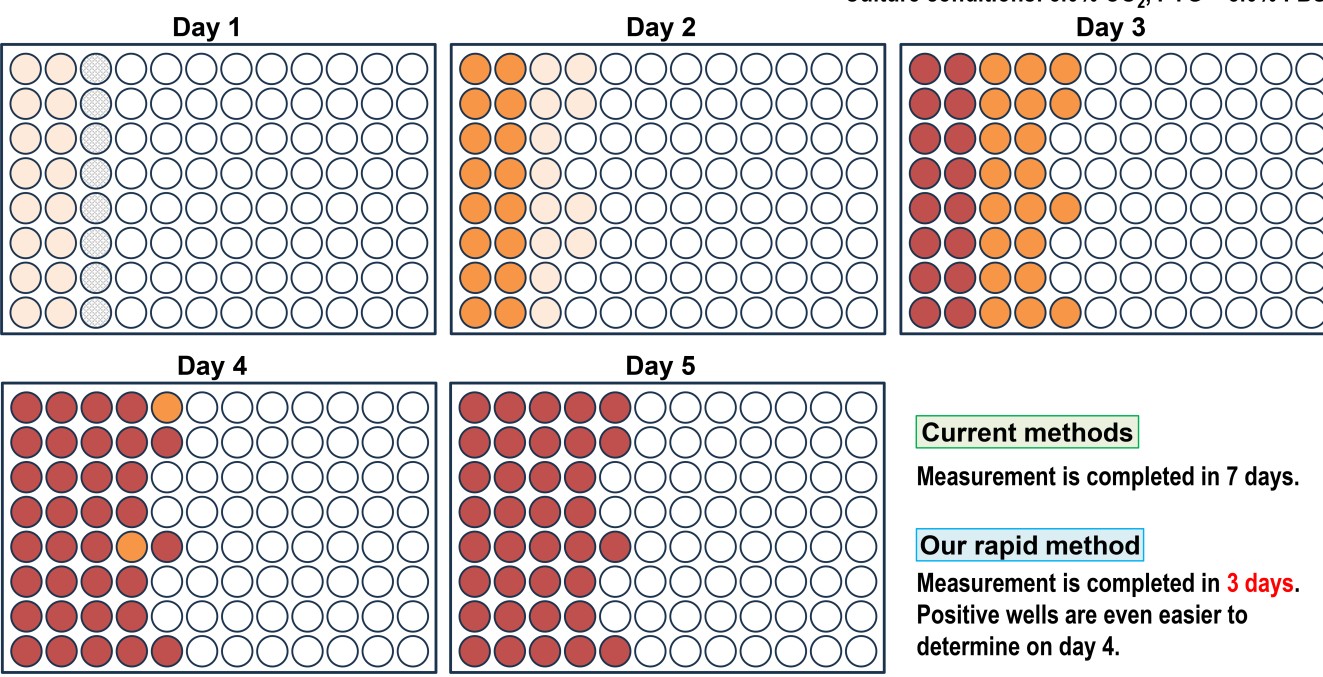

FIG 1 Overview of living *Acanthamoeba castellanii* cyst count (LACC) measurement. AC cyst samples were serially diluted and inoculated onto a 96-well plate at 100 µL per well. Following this, each well was examined under an inverted microscope on days 1 through 7 of incubation to determine whether cysts had excysted. Wells in which cysts had excysted or cysts had excysted and trophozoites proliferated were considered positive. Positive wells were classified into three stages: stage 1: excystation observed in some cysts. Stage 2: proliferation of trophozoites, along with excystation, observed. Stage 3: multiple proliferating trophozoites observed. Although positive wells are relatively difficult to determine at stage 1, they become increasingly easy to distinguish as they progress to stage 2 and then stage 3. We developed a rapid protocol to accurately quantify living AC cysts that could be completed within only 3 days. Moreover, positive wells were considerably easier to distinguish on day 4.

significantly promoted when cultured with 5.0% $CO_2$, with approximately half of the cysts excysted. Furthermore, the addition of 5% FBS to the PYG medium induced excystation of almost all cysts (Fig. 2A).

On day 3 of incubation, some cysts had excysted in culture with 0.04% $CO_2$, whereas the majority of cysts were excysted when cultured in PYG medium supplemented with 5% FBS. Moreover, nearly all cysts were excysted in culture with 5.0% $CO_2$, and active trophozoite proliferation was observed. Furthermore, the addition of 5% FBS to the PYG medium further accelerated the proliferation rate, and numerous excysted and proliferating trophozoites were observed (Fig. 2B).

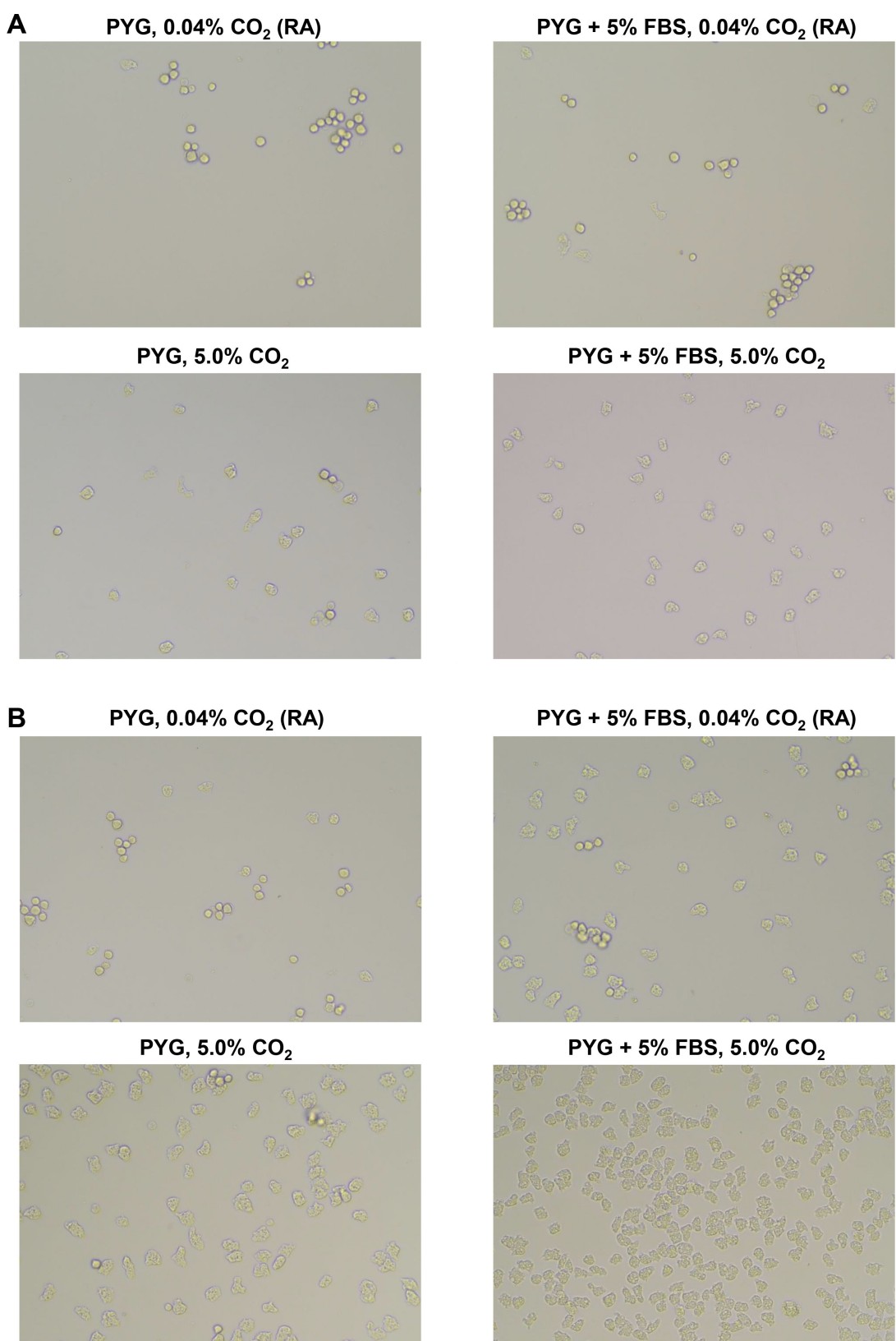

**FIG 2** Observation of excystation under various culture conditions. PYG medium containing AC standard strain (ATCC 50492) cysts ($1 \times 10^3$ /mL) was inoculated into 96-well plates. The FBS concentration in the PYG medium and $CO_2$ concentration during culture varied, and the excystation of AC cysts was observed on days 2 (A) and 3 (B) of incubation under an inverted microscope.

## Measurement of living AC cyst counts (LACC)

In the high-concentration sample (5 $Log_{10}$ LACC/mL) of AC standard strain cysts, the measured LACC plateaued on day 6 (peaked on day 6 and remained constant thereafter) under 0.04% $CO_2$ culture conditions. In contrast, under 2.5% or 5.0% $CO_2$ culture conditions, the LACC increased daily before plateauing on day 4. A similar trend was observed in the low-concentration sample (3 $Log_{10}$ LACC/mL). Under 0.04% $CO_2$ culture conditions, the LACC plateaued on day 6, whereas under 2.5% or 5.0% $CO_2$ culture conditions, it plateaued on day 4 (Fig. 3A; Table 1). Furthermore, the LACC increased daily when FBS was added to the PYG medium. Specifically, when 5% or more FBS was added to the culture, the LACC plateaued on day 5 under 0.04% $CO_2$ culture conditions and on day 3 under 2.5% or 5.0% $CO_2$ culture conditions (Fig. 3B through D; Table 1).

For the high- and low-concentration samples of the AC-W clinical strain cysts, the LACC plateaued on day 5 under 0.04% $CO_2$ culture conditions. In contrast, under 2.5% or 5.0% $CO_2$ culture conditions, the LACC increased daily before plateauing on day 4 (Fig. 4A; Table 1). Furthermore, when 5% or more FBS was added to the PYG medium, the LACC plateaued on day 3 under 2.5% or 5.0% $CO_2$ culture conditions (Fig. 4B through D; Table 1).

Similarly, for high- and low-concentration samples of AC-U clinical strain cysts, the LACC plateaued on days 5 or 6 under 0.04% $CO_2$ culture conditions. In contrast, under 2.5% or 5.0% $CO_2$ culture conditions, the LACC increased daily before plateauing on day 4 (Fig. 5A; Table 1). However, when 5% or more FBS was added to the PYG medium, the LACC plateaued on day 3 under 2.5% or 5.0% $CO_2$ culture conditions (Fig. 5B through D; Table 1).

Taken together, these results demonstrate that for all evaluated AC strain cysts, the LACC plateaued on day 3 when the PYG medium was supplemented with 5% or more FBS under 2.5% or higher $CO_2$ culture conditions; thus, the LACC could be measured within 3 days (Table 1).

## Evaluation of the accuracy of the LACC assay

The accuracy of the LACC assay was determined by analyzing the discrepancy between the theoretical and actual measured values obtained using various protocols. For all protocols, the mean "count divergence of the LACC assay" was below 0.15, and the 95% confidence interval was within the range of −0.5 to 0.5 (Table 2). This indicates that the LACC assay had high accuracy across all protocols, including the protocol developed in this study.

## The accuracy of the LACC assay using the new rapid protocol for AC cysts exposed to disinfectants

The accuracy of the LACC assay using the new rapid protocol was evaluated by comparing its measurements with those obtained using the current 7 day protocol. For all chemically treated samples, the mean difference in logarithmic LACC was below 0.10, with a 95% confidence interval of −0.25 to 0.25 (Table 3). These results indicate that the new rapid protocol provides high accuracy for the LACC assay in chemically treated samples.

## Changes in PYG medium pH over time

Changes in the pH of the PYG medium during incubation were measured under 0.04% and 5.0% $CO_2$ culture conditions. Over 7 days, the pH was maintained in the range of 7.0–7.1 and 6.8–6.9 under 0.04% $CO_2$ and 5.0% $CO_2$ culture conditions, respectively (Table 4).

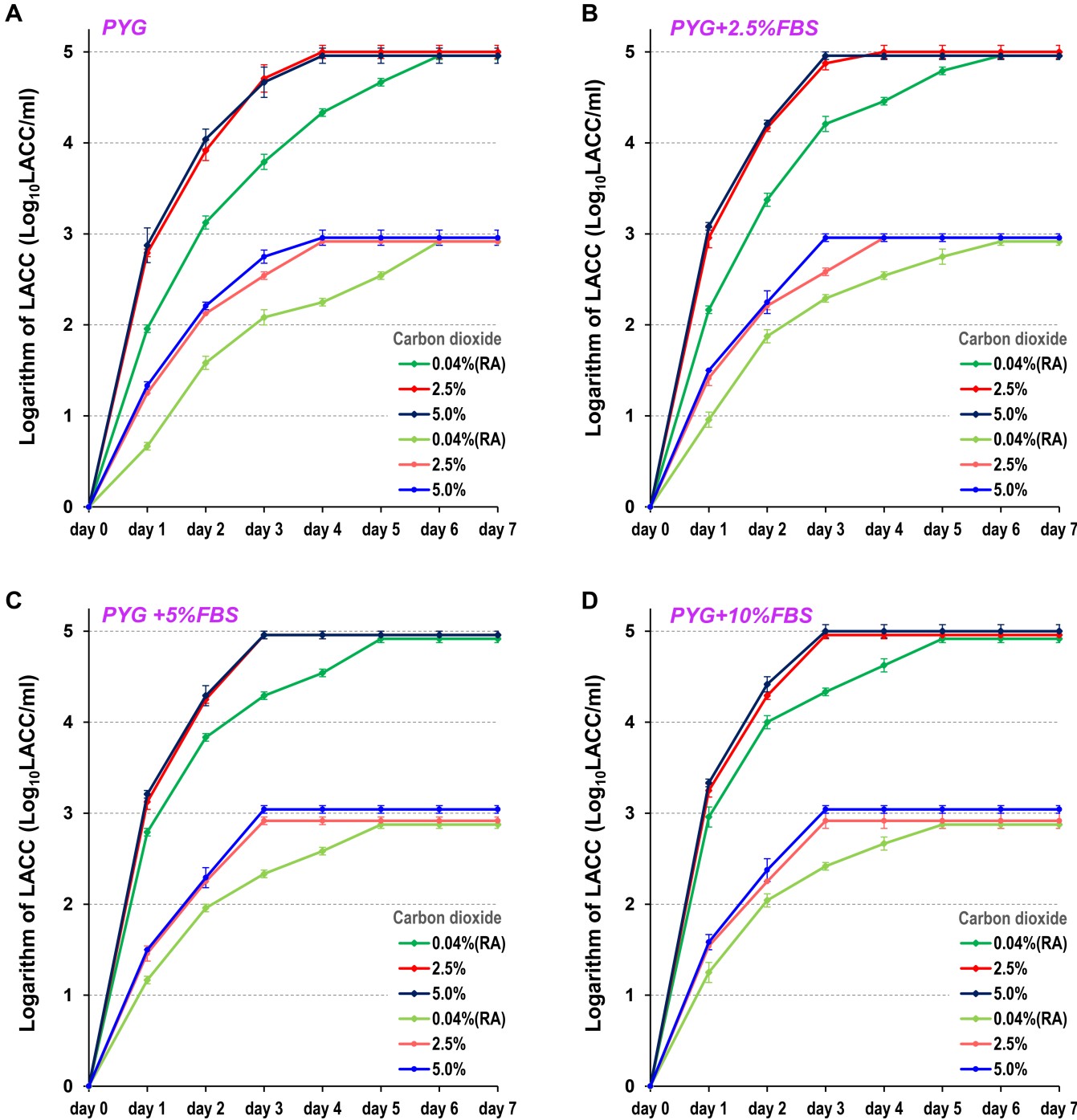

**FIG 3** LACC measurements of standard strain (ATCC 50492) cysts using various protocols. LACC was measured in samples adjusted to concentrations of 5 $Log_{10}$ LACC/mL or 3 $Log_{10}$ LACC/mL. LACC was measured after 1 to 7 days of incubation across 12 protocols, combining four FBS concentrations (0% [A], 2.5% [B], 5.0% [C], and 10% [D]) added to PYG medium and three $CO_2$ concentrations (0.04%, 2.5%, and 5.0%) in the incubator. The 0.04% $CO_2$ condition corresponds to a room air (RA) environment. For each measurement, three independent experiments were performed, and the results are expressed as the mean ± standard error of the mean.

## DISCUSSION

The current method for quantifying living AC cysts has high measurement accuracy and is suitable for evaluating cyst disinfection as well as the environmental stability of cysts. The LACC is determined using this method through the direct observation of

**TABLE 1** Time required for the LACC to reach the upper limit[a]

| | | Time required for LACC to reach the upper limit (day) | | | | | | | |
|---|---|---|---|---|---|---|---|---|---|
| | | $5 \, Log_{10}$ LACC/mL | | | | $3 \, Log_{10}$ LACC/mL | | | |
| | | PYG | PYG + 2.5% FBS | PYG + 5% FBS | PYG + 10% FBS | PYG | PYG + 2.5% FBS | PYG + 5% FBS | PYG + 10% FBS |
| ATCC 50492 | 0.04% $CO_2$ (RA) | 6 | 6 | 5 | 5 | 6 | 6 | 5 | 5 |
| | 2.5% $CO_2$ | 4 | 4 | 3 | 3 | 4 | 4 | 3 | 3 |
| | 5.0% $CO_2$ | 4 | 3 | 3 | 3 | 4 | 3 | 3 | 3 |
| AC-W (clinical strain) | 0.04% $CO_2$ (RA) | 5 | 5 | 5 | 5 | 5 | 5 | 5 | 5 |
| | 2.5% $CO_2$ | 4 | 4 | 3 | 3 | 4 | 4 | 3 | 3 |
| | 5.0% $CO_2$ | 4 | 3 | 3 | 3 | 4 | 3 | 3 | 3 |
| AC-U (clinical strain) | 0.04% $CO_2$ (RA) | 6 | 5 | 5 | 5 | 5 | 5 | 5 | 5 |
| | 2.5% $CO_2$ | 4 | 4 | 3 | 3 | 4 | 4 | 3 | 3 |
| | 5.0% $CO_2$ | 4 | 4 | 3 | 3 | 4 | 4 | 3 | 3 |

[a]LACC was measured in samples adjusted to concentrations of $5 \, Log_{10}$ LACC/mL or $3 \, Log_{10}$ LACC/mL. LACC was measured after 1 to 7 days of incubation across 12 protocols, combining four FBS concentrations (0%, 2.5%, 5.0%, and 10%) added to PYG medium and three $CO_2$ concentrations (0.04%, 2.5%, and 5.0%) in the incubator. From these measurement data, the time required for LACC to reach the upper limit was calculated.

cyst excystation and trophozoite proliferation in individual wells under a microscope (18–20). Although the LACC is calculated as a logarithmic value using this method, logarithmic data are well-suited for comparing and analyzing disinfection effectiveness and environmental stability. In this study, it was first demonstrated that this quantification method is accurate, with minimal divergence from theoretical values. However, this current quantification method requires 7 days to obtain results, which has been a long-standing drawback (18–20). In this study, we overcame this limitation and developed a novel, rapid protocol for accurately quantifying living AC cysts that significantly reduced the time needed for analysis (Fig. 1).

Previously, Khan et al. reported that excystation of AC cysts may be accelerated in the presence of FBS and $CO_2$ (26). In this present study, we independently verified that excystation was enhanced under these conditions. Building on this, we identified the culture conditions under which the quantification of living AC cysts can be completed in the shortest time. The excystation of AC cysts and proliferation of trophozoites were promoted by increasing the FBS concentration in the PYG medium and increasing the $CO_2$ concentration during culture. This significantly reduced the time needed for the daily LACC values to plateau, indicating the time required to complete the quantification of living AC cysts. Ultimately, our investigation revealed that LACC measurements can be completed within 3 days for standard and clinical strain cysts by supplementing PYG medium with 5% or more FBS and performing incubation with 2.5% or higher $CO_2$. Furthermore, we demonstrated that this new rapid protocol can quantify living AC cysts with great accuracy, exhibiting the same accuracy as the current 7 day protocol. In additional analyses, although chemical treatment (i.e., exposure to benzalkonium chloride [BAC]) reduced LACC, there was no difference between LACC values obtained using the new rapid protocol and those obtained using the current 7 day protocol. These findings suggest that the new rapid protocol provides high accuracy for the LACC assay in chemically treated samples.

However, one limitation of this study is that the mechanism through which the excystation of AC cysts is promoted by increasing $CO_2$ concentrations during culture is still unclear. Typically, when culturing cell lines under 5.0% $CO_2$ conditions, $CO_2$ plays a key role in adjusting and maintaining the pH of the culture medium (27). We therefore assessed changes in the pH of the PYG medium over time under 0.04% and 5.0% $CO_2$ culture conditions. Although the pH under 5.0% $CO_2$ culture conditions was slightly lower than that under 0.04% $CO_2$ conditions, the pH remained within the optimal range for AC over 7 days under both conditions. Therefore, we concluded that the pH of the culture medium does not play a direct role in promoting excystation by $CO_2$. Khan et al. proposed that the promotion of excystation in the presence of $CO_2$ may be attributable

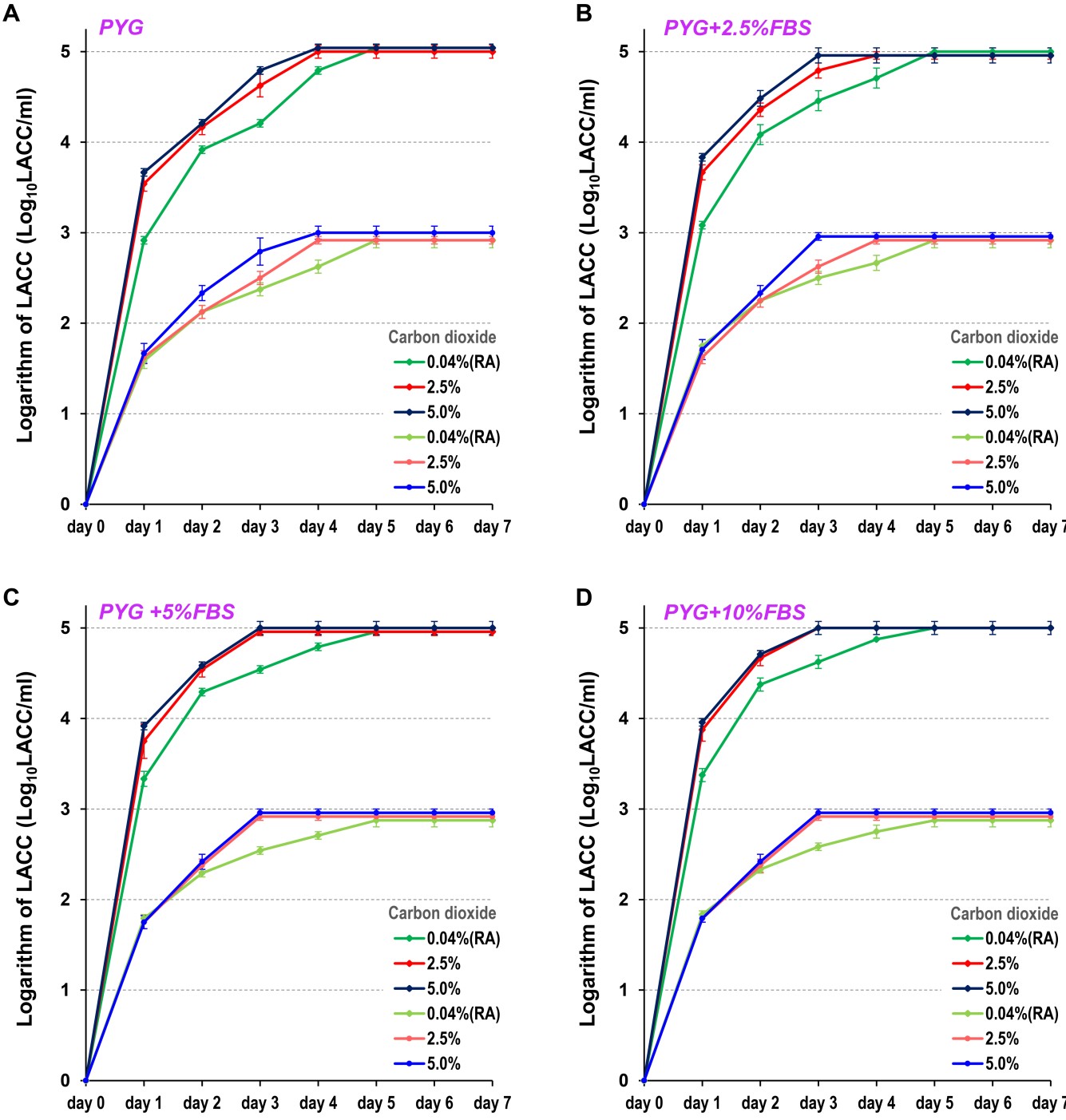

**FIG 4** LACC measurements of the clinical strain AC-W using various protocols. LACC was measured in samples adjusted to concentrations of 5 Log10 LACC/mL or 3 Log10 LACC/mL. LACC was measured after 1 to 7 days of incubation across 12 protocols, combining four FBS concentrations (0% [A], 2.5% [B], 5.0% [C], and 10% [D]) added to PYG medium and three CO2 concentrations (0.04%, 2.5%, and 5.0%) in the incubator. The 0.04% CO2 condition corresponds to a room air (RA) environment. For each measurement, three independent experiments were performed, and the results are expressed as the mean ± standard error of the mean.

to the similarity of the conditions to the high $CO_2$ pressure conditions in human tissues, and we currently support their hypothesis (26). Future studies should investigate the mechanisms through which increasing $CO_2$ concentrations during culture promote the excystation of AC cysts.

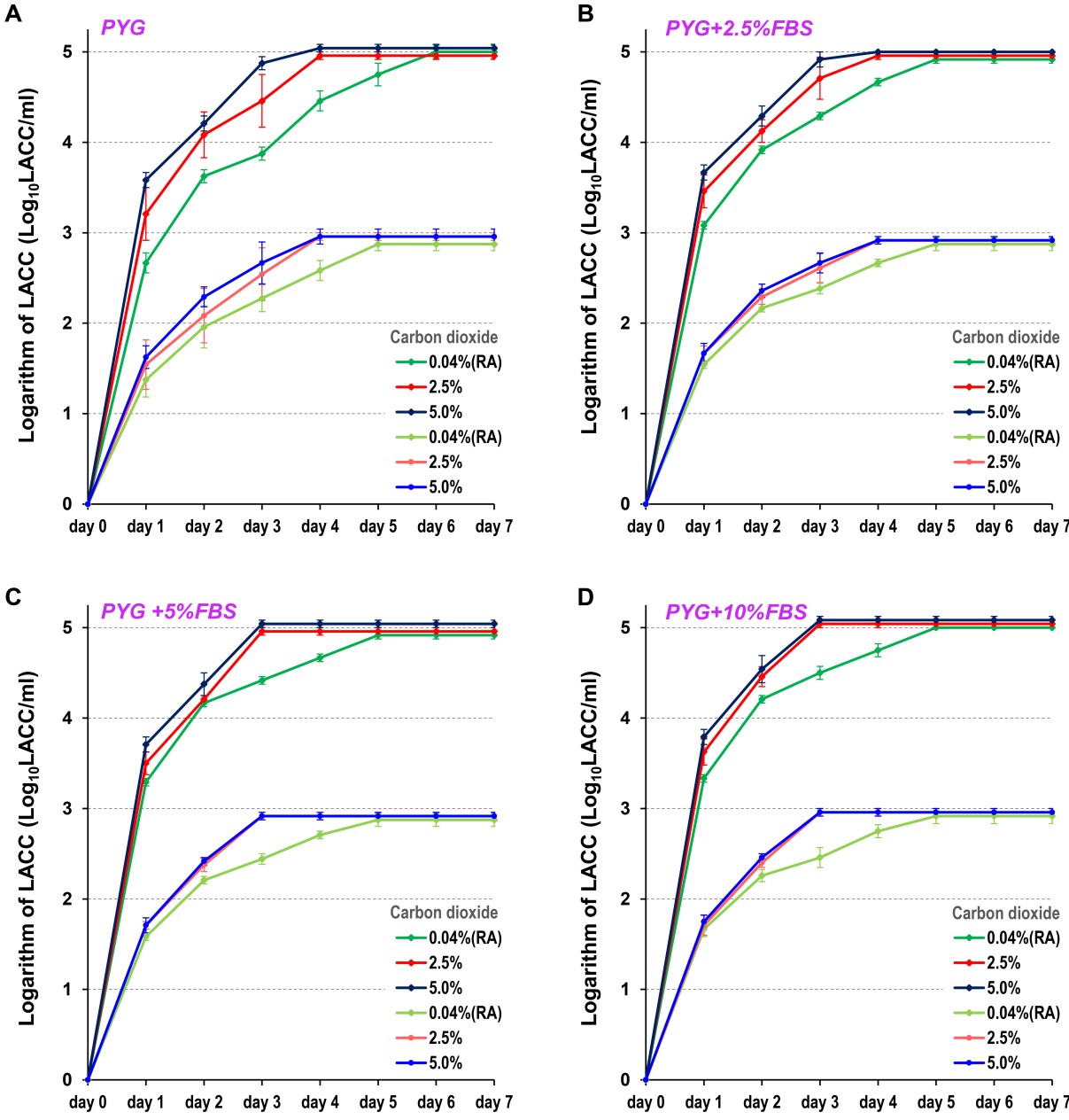

**FIG 5** LACC measurements of the clinical strain AC-U using various protocols. LACC was measured in samples adjusted to concentrations of 5 Log10 LACC/mL or 3 Log10 LACC/mL. LACC was measured after 1 to 7 days of incubation across 12 protocols, combining four FBS concentrations (0% [A], 2.5% [B], 5.0% [C], and 10% [D]) added to PYG medium and three CO2 concentrations (0.04%, 2.5%, and 5.0%) in the incubator. The 0.04% CO2 condition corresponds to a room air (RA) environment. For each measurement, three independent experiments were performed, and the results are expressed as the mean ± standard error of the mean.

In conclusion, the accurate method for quantifying living AC cysts developed in this study, which can be completed in 3 days, can reduce the time required for measurement by more than half compared to current methods and holds potential for expediting the evaluation of the effectiveness of disinfectants and other drugs against AC cysts. In addition, this quantitative method can also be applied to quantify living cysts of other amoebas, potentially reducing the time required for measurement.

TABLE 2 Count divergence of the LACC assay for three *Acanthamoeba* strains under various culture conditions[a]

| | | Count divergence of logarithmic LACC, mean (95% confidence interval) | | | | | | | |
| | | 5 Log$_{10}$ LACC/mL | | | | 3 Log$_{10}$ LACC/mL | | | |
| | | PYG | PYG + 2.5% FBS | PYG + 5% FBS | PYG + 10% FBS | PYG | PYG + 2.5% FBS | PYG + 5% FBS | PYG + 10% FBS |
|---|---|---|---|---|---|---|---|---|---|
| ATCC 50492 | 0.04% CO$_2$ (RA) | −0.042 (−0.124–0.040) | −0.042 (−0.124–0.040) | −0.083 (−0.165–0.001) | −0.083 (−0.165–0.001) | −0.083 (−0.246–0.080) | −0.083 (−0.246–0.080) | −0.125 (−0.266–0.016) | −0.125 (−0.266–0.016) |
| | 2.5% CO$_2$ | 0.000 (−0.141–0.141) | 0.000 (−0.141–0.141) | −0.042 (−0.124–0.040) | −0.042 (−0.124–0.040) | −0.083 (−0.165–0.001) | −0.042 (−0.124–0.040) | −0.083 (−0.165–0.001) | −0.083 (−0.246–0.080) |
| | 5.0% CO$_2$ | −0.042 (−0.205–0.121) | −0.042 (−0.205–0.121) | −0.042 (−0.124–0.040) | 0.000 (−0.141–0.141) | −0.042 (−0.205–0.121) | −0.042 (−0.124–0.040) | 0.042 (−0.040–0.124) | 0.042 (−0.040–0.124) |
| AC-W (clinical strain) | 0.04% CO$_2$ (RA) | 0.042 (−0.040–0.124) | 0.000 (0.000–0.000) | −0.042 (−0.124–0.040) | 0.000 (0.000–0.000) | −0.083 (−0.246–0.080) | −0.083 (−0.246–0.080) | −0.125 (−0.266–0.016) | −0.125 (−0.266–0.016) |
| | 2.5% CO$_2$ | 0.000 (−0.141–0.141) | −0.042 (−0.124–0.040) | −0.042 (−0.124–0.040) | 0.000 (0.000–0.000) | −0.083 (−0.165–0.001) | −0.083 (−0.165–0.001) | −0.083 (−0.165–0.001) | −0.083 (−0.165–0.001) |
| | 5.0% CO$_2$ | 0.042 (−0.040–0.124) | −0.042 (−0.205–0.121) | 0.000 (−0.141–0.141) | 0.000 (0.000–0.000) | 0.000 (−0.141–0.141) | −0.042 (−0.124–0.040) | −0.042 (−0.124–0.040) | −0.042 (−0.124–0.040) |
| AC-U (clinical strain) | 0.04% CO$_2$ (RA) | 0.000 (−0.141–0.141) | −0.083 (−0.165–0.001) | −0.083 (−0.165–0.001) | 0.000 (0.000–0.000) | −0.125 (−0.266–0.016) | −0.125 (−0.266–0.016) | −0.125 (−0.266–0.016) | −0.083 (−0.246–0.080) |
| | 2.5% CO$_2$ | −0.042 (−0.124–0.040) | −0.042 (−0.124–0.040) | −0.042 (−0.124–0.040) | 0.042 (−0.040–0.124) | −0.042 (−0.124–0.040) | −0.083 (−0.165–0.001) | −0.083 (−0.165–0.001) | −0.042 (−0.124–0.040) |
| | 5.0% CO$_2$ | 0.042 (−0.040–0.124) | 0.000 (0.000–0.000) | 0.042 (−0.040–0.124) | 0.083 (0.001–0.165) | −0.042 (−0.205–0.121) | −0.083 (−0.165–0.001) | −0.083 (−0.165–0.001) | −0.042 (−0.124–0.040) |

[a]LACC was measured in samples adjusted to concentrations of 5 Log$_{10}$ LACC/mL or 3 Log$_{10}$ LACC/mL. LACC was measured after 1 to 7 days of incubation across 12 protocols, combining four FBS concentrations (0%, 2.5%, 5.0%, and 10%) added to PYG medium and three CO$_2$ concentrations (0.04%, 2.5%, and 5.0%) in the incubator. From these measurement data, the count divergence of logarithmic LACC, which represents the accuracy of the LACC assay for each protocol, was calculated.

**TABLE 3** Accuracy of the LACC assay using the new rapid protocol for three *Acanthamoeba* strains after chemical treatment[d]

| Chemical treatments | Protocol | Logarithmic LACC (Log$_{10}$ LACC/ml), mean ± SE | | | Difference in logarithmic LACC,[c] mean (95% confidence interval) | | |
|---|---|---|---|---|---|---|---|
| | | ATCC 50492 | AC-W | AC-U | ATCC 50492 | AC-W | AC-U |
| 10 ppm BAC, 1 h | Current 7 day protocol[a] | 3.92 ± 0.08 | 3.75 ± 0.14 | 3.79 ± 0.11 | | | |
| | New rapid protocol[b] | 3.96 ± 0.04 | 3.80 ± 0.11 | 3.75 ± 0.14 | 0.042 (−0.174–0.258) | 0.042 (−0.040–0.123) | −0.042 (−0.123–0.040) |
| 10 ppm BAC, 6 h | Current 7 day protocol[a] | 3.38 ± 0.08 | 3.33 ± 0.08 | 3.38 ± 0.07 | | | |
| | New rapid protocol[b] | 3.35 ± 0.07 | 3.33 ± 0.04 | 3.33 ± 0.11 | −0.042 (−0.123–0.040) | 0.000 (−0.141–0.141) | −0.042 (−0.205–0.122) |
| 100 ppm BAC, 1 h | Current 7 day protocol[a] | 0.78 ± 0.04 | 0.71 ± 0.11 | 0.67 ± 0.08 | | | |
| | New rapid protocol[b] | 0.75 ± 0.07 | 0.75 ± 0.00 | 0.63 ± 0.07 | −0.083 (−0.165–0.002) | 0.042 (−0.174–0.258) | −0.042 (−0.123–0.040) |
| 100 ppm BAC, 6 h | Current 7 day protocol[a] | UD[e] | UD | UD | | | |
| | New rapid protocol[b] | UD | UD | UD | | | |

[a]In the current 7 day protocol, LACC was measured after incubation for 7 days in PYG medium without FBS under 0.04% $CO_2$ culture conditions.
[b]In the new rapid protocol, LACC was measured after incubation for 3 days in PYG medium containing 5% FBS under 2.5% $CO_2$ culture conditions.
[c]The difference between the logarithmic LACC values obtained using the new rapid protocol and the current 7 day protocol indicates the accuracy of the LACC assay using the new rapid protocol.
[d]High-concentration samples of the AC standard strain cysts, AC-W clinical strain cysts, and AC-U clinical strain cysts (5 Log$_{10}$ LACC/mL) were exposed to 10 or 100 ppm benzalkonium chloride (BAC) for 1 or 6 h, after which LACC was measured.
[e]UD, under the detection limit, that is, 0.5 Log$_{10}$LACC/ml or less.

## MATERIALS AND METHODS

### AC strains and culture method

Standard AC strains belonging to the T4 genotype (ATCC 50492) were purchased from the American Type Culture Collection (ATCC, VA, USA). Two clinical AC strains belonging to the T4 genotype were isolated from patients with AK and labeled AC-U and AC-W.

These AC trophozoites were adapted to grow in 25 cm$^2$ tissue culture flasks in 4 mL of PYG medium containing 2.0% proteose peptone (Thermo Fisher Scientific, MA, USA), 0.1% yeast extract (Solabia Biokar Diagnostics, Pantin, France), and 1.8% glucose (Nacalai Tesque, Kyoto, Japan) without shaking at 30°C (20, 28). Prior to the start of this study, PYG medium containing the cultured AC trophozoites was centrifuged at 5,000 *g* for 10 min, and the pellet was plated on brain heart infusion agar to screen for bacterial contamination.

### Preparation of AC cysts

Cysts were prepared from late log phase trophozoite cultures. Specifically, the trophozoites were transferred from PYG medium to Neff's encystment medium containing 0.1 M KCl, 8 mM MgSO$_4$·7H$_2$O, 0.4 mM CaCl$_2$·2H$_2$O, 1 mM NaHCO$_3$, and 10 mM tris(hydroxymethyl)aminomethane and cultured for 6–7 days to obtain mature cysts (5). Following this, any remaining trophozoites were solubilized by treatment with 0.5% sodium dodecyl sulfate (FUJIFILM Wako Pure Chemical Corporation, Osaka, Japan) at room temperature for 10 min (4, 29–31). The mature cysts were then harvested and washed twice with 8 mL of phosphate-buffered saline (Nacalai Tesque).

**TABLE 4** Changes in the pH of PYG medium over time in the presence of amoeba cysts

| | Day 0 | Day 1 | Day 2 | Day 3 | Day 4 | Day 5 | Day 6 | Day 7 |
|---|---|---|---|---|---|---|---|---|
| 0.04% $CO_2$ (RA) | 7.04 ± 0.02 | 7.01 ± 0.06 | 7.02 ± 0.02 | 7.05 ± 0.02 | 7.04 ± 0.02 | 7.03 ± 0.01 | 7.03 ± 0.01 | 7.05 ± 0.01 |
| 5.0% $CO_2$ | 6.89 ± 0.03 | 6.85 ± 0.02 | 6.85 ± 0.02 | 6.85 ± 0.02 | 6.84 ± 0.02 | 6.84 ± 0.01 | 6.82 ± 0.02 | 6.82 ± 0.01 |

## Observation of excystation under various culture conditions

The prepared AC cysts were adjusted to a concentration of $1 \times 10^3$ /mL in PYG medium using a hemocytometer, and 100 mµL of the PYG medium containing AC cysts was inoculated into 96-well plates. The FBS concentration in the PYG medium and the $CO_2$ concentration varied during the culture, and the excystation of AC cysts was observed daily for 3 days using an inverted microscope. Briefly, culture was performed under four conditions: two with 0% and 5% FBS added to the PYG medium, and two with $CO_2$ concentrations of 0.04% (corresponding to room air, i.e., atmospheric conditions) and 5.0%.

## Measurement of LACC using various protocols

The prepared AC cysts were adjusted to concentrations of $1 \times 10^5$ /mL or $1 \times 10^3$ /mL in PYG medium, corresponding to 5 $Log_{10}$ LACC/mL or 3 $Log_{10}$ LACC/mL, respectively, using a hemocytometer. LACC was measured after 1 to 7 days of incubation under each culture condition. Specifically, the prepared AC cyst sample was serially diluted and inoculated onto a 96-well plate at 100 µL per well. Each well was examined daily on days 1 through 7 of incubation under an inverted microscope to determine whether cysts had excysted (Fig. 1). Wells in which cysts had excysted or in which cysts had excysted and trophozoites had proliferated were determined as positive. LACC was calculated using the Spearman-Karber method based on data from positive wells in each dilution series (18–20). A total of 12 culture protocols were used to measure LACC, combining four FBS concentrations (0%, 2.5%, 5.0%, and 10%) in the PYG medium and three $CO_2$ concentrations in the incubator (0.04%, 2.5%, and 5.0%). For each measurement, three independent experiments were performed, and the results are expressed as the mean ± standard error of the mean.

## Measurement of LACC in chemically treated samples

The prepared AC cysts were adjusted to concentrations of $1 \times 10^5$ /mL in PYG medium, corresponding to 5 $Log_{10}$ LACC/mL, using a hemocytometer. These high-concentration samples were exposed to 10 or 100 ppm BAC for 1 or 6 h, after which LACC was measured using two protocols: the current 7 day protocol and the new rapid protocol. In the current 7 day protocol, LACC was measured after incubation for 7 days in PYG medium without FBS under 0.04% $CO_2$ culture conditions. In the new rapid protocol, LACC was measured after incubation for 3 days in PYG medium containing 5% FBS under 2.5% $CO_2$ culture conditions. For each measurement, three independent experiments were performed, and the results are expressed as the mean ± standard error of the mean. In addition, we calculated the difference between the logarithmic LACC obtained using the new rapid protocol and the current 7 day protocol to indicate the accuracy of the LACC assay using the new rapid protocol.

## Measurement of PYG medium pH over time

The pH of PYG medium without FBS was measured during AC cyst culture over 7 days in an incubator with a $CO_2$ concentration of 0.04% or 5.0%. For each measurement, three independent experiments were performed, and the results were expressed as the mean ± standard error of the mean.

## Statistical analysis

The theoretical LACC values for the two concentration-adjusted AC cyst samples used in this study were set to either 5.0 $Log_{10}$ LACC/mL or 3.0 $Log_{10}$ LACC/mL. During LACC measurements over time, the values peaked at the 7 day measurement and then plateaued. This upper limit was defined as the actual measurement value for each protocol. The difference between the logarithmic theoretical value and the logarithmic actual measured value was defined as the "count divergence of the LACC assay," which

represents the accuracy of the LACC assay for each protocol. Specifically, a protocol was deemed to accurately measure the LACC if the 95% confidence interval of the count divergence of the LACC assay fell within the range of −0.5 to 0.5 (32). Statistical analysis was performed using GraphPad Prism 7 software (GraphPad, La Jolla, CA, USA).

## ACKNOWLEDGMENTS

We thank Editage for English language editing.

This research was supported by the Lotte Research Promotion Grant, Takahashi Industrial Economic Research Foundation, and Takeda Science Foundation.

Study concept and design: R.H. Data acquisition: R.H., K.M., N.H., T.M., T.K., and A.S. Data analysis and interpretation: R.H., K.M., M.Y., and T.N. Drafting of the manuscript: R.H. Statistical analysis: R.H. and K.M. Secured funding: R.H. and T.N. Administrative/technical/material support: R.H. and M.Y. Study supervision: R.H., M.Y., and T.N.

## AUTHOR AFFILIATIONS

[1]Department of Infectious Diseases, Graduate School of Medical Science, Kyoto Prefectural University of Medicine, Kyoto, Japan

[2]Department of Molecular Gastroenterology and Hepatology, Graduate School of Medical Science, Kyoto Prefectural University of Medicine, Kyoto, Japan

[3]Department of Forensic Medicine, Graduate School of Medical Science, Kyoto Prefectural University of Medicine, Kyoto, Japan

## AUTHOR ORCIDs

Ryohei Hirose  http://orcid.org/0000-0003-0418-1390

## FUNDING

| Funder | Grant(s) | Author(s) |
| --- | --- | --- |
| Lotte Research Promotion Grant | | Ryohei Hirose |
| Takahashi Industrial and Economic Research Foundation | | Ryohei Hirose |
| Takeda Science Foundation | | Ryohei Hirose |

## AUTHOR CONTRIBUTIONS

Kazushi Matsubara, Data curation, Formal analysis, Investigation, Methodology | Ryohei Hirose, Conceptualization, Data curation, Formal analysis, Funding acquisition, Investigation, Methodology, Project administration, Resources, Software, Supervision, Validation, Visualization, Writing – original draft, Writing – review and editing | Norihide Hasegawa, Data curation | Takumi Minamiyama, Data curation | Taku Kano, Data curation | Akinobu Sai, Data curation | Minoru Yamada, Methodology, Resources | Takaaki Nakaya, Conceptualization, Project administration, Resources, Supervision, Validation

## DATA AVAILABILITY

All data included in this study are available from the corresponding author on request.

## ADDITIONAL FILES

The following material is available online.

Open Peer Review

**PEER REVIEW HISTORY (review-history.pdf).** An accounting of the reviewer comments and feedback.

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
