## [Reviewer comments · Microbiology Spectrum]

Microbiology Spectrum

A rapid and accurate protocol for quantifying living *Acanthamoeba castellanii* cysts

Kazushi Matsubara, Ryohei Hirose, Hirohide Hasegawa, Takumi Minamiyama, Taku Kano, Akinobu Sai, Minoru Yamada, and Takaaki Nakaya

Corresponding Author(s): Ryohei Hirose, Kyoto Furitsu Ika Daigaku

Review Timeline:

Submission Date:	April 16, 2025
Editorial Decision:	July 26, 2025
Revision Received:	August 14, 2025
Accepted:	August 25, 2025

Editor: Michael Ginger

Reviewer(s): The reviewers have opted to remain anonymous.

Transaction Report:

DOI: <https://doi.org/10.1128/spectrum.01170-25>

Re: Spectrum01170-25 (A rapid and accurate method for quantifying living *Acanthamoeba castellanii* cysts)

Dear Dr. Ryohei Hirose:

Thank you for the submission of your manuscript to ASM Microbiology Spectrum. I apologise for the time taken to collect reviewers' comments. Both reviewers have extensive expertise in the biology of cyst-forming amoebae. Each reviewer is also supportive of the protocol development you report, but each feels that presently the manuscript lacks a little depth. Here, I invite you to submit a revised manuscript to the journal. To an extent whether you choose to add further data along the lines encouraged by the reviewers is up to you; I feel their suggestions for additional experiments or data are reasonable. However, what I will insist on is that you pay very careful attention to technical points 1) and 2) raised by reviewer #2. Satisfactory revision of your manuscript should see it acceptable for publication in the journal. Below you will find instructions from the Spectrum editorial office, and the reviewer comments.

Revision Guidelines

Sincerely,
Michael Ginger
Editor
Microbiology Spectrum

Reviewer #1 (Comments for the Author):

The authors show that excystation of *Acanthamoeba* cysts and release of trophozoites increases when cysts are cultured in

medium with 5% fetal bovine serum and 5% CO₂, so that viability of cysts can be judged by inverted phase microscopy in just three days rather than one week. They suggest but do not demonstrate that this would be a great way to assay the effect of various chemical treatments on cyst viability. While it is nice that they assayed cysts from ATCC as well as from two patients with AK, Figs 3 to 5 are essentially identical and could easily be combined.

Reviewer #2 (Comments for the Author):

This study describes a useful tool to evaluate the viability of *Acanthamoeba castellanii* cysts through the excystation method. The conclusions highlight that the addition of FBS and 5%CO₂ accelerate this process and could be used to determine whether cysts are viable after exposure to anti-amoebic compounds. The study only describes one species and strain of *Acanthamoeba* and it would be useful to compare different species and strains as it is known that different species have different excystation periods. In addition, the study would benefit from a cyst disinfection/anti-amoebic assay to determine if the assay can be used to identify live/dead cysts. In addition, there are a couple of technical queries:

1. the encystation method described specifies that the trophozoites are exposed to Neff encystation medium for 6-7 days until cysts are observed and then these were harvested. How do the authors know that the cysts were mature? Typically, the cysts are washed in 0.5% SDS to verify maturity. This is not documented in the methodology section.
2. From the microscopy images, it looks like growth of trophozoites is also accelerated in FBS and 5% CO₂. How do the authors account for this?
3. Typically, it is expected that ATCC strains are axenic, but some bacteria can be present. Did the authors perform any screening for bacterial presence prior to the experiments?
4. line 43 page 8 what does plateau mean, upper limit of what?

Other comments:

1. line 79 page 5: use the word 'protist' instead of 'protozoan'
2. line 85 page 5: instead of 'pathogenic' use 'an opportunistic pathogen'. It can also cause other diseases. Not just AK.
3. In the figures it is LLAC and in text it is LAAC.

Response to Reviewer Comments

We greatly appreciate your kind review and have responded to the reviewers' comments in as much detail as possible.

Reviewer #1 (Comments for the Author):

The authors show that excystation of *Acanthamoeba* cysts and release of trophozoites increases when cysts are cultured in medium with 5% fetal bovine serum and 5% CO₂, so that viability of cysts can be judged by inverted phase microscopy in just three days rather than one week. They suggest but do not demonstrate that this would be a great way to assay the effect of various chemical treatments on cyst viability. While it is nice that they assayed cysts from ATCC as well as from two patients with AK, Figs 3 to 5 are essentially identical and could easily be combined.

Response: We truly appreciate your helpful comments. In response, as an additional analysis, we evaluated the accuracy of the LACC assay using the new rapid protocol for AC cysts after chemical treatment. A new table has been created accordingly, and additional information has been added to the Materials and Methods, Results, and Discussion sections. Although chemical treatment (i.e., exposure to benzalkonium chloride) reduced LACC as expected, there was no difference between the values obtained using the new rapid protocol and the 7-day protocol. Thus, the new rapid protocol provides high accuracy for the LACC assay in chemically treated samples. We also believe that presenting the data in three figures (Figures 3–5) makes it easier for readers to understand and more useful.

Reviewer #2 (Comments for the Author):

This study describes a useful tool to evaluate the viability of *Acanthamoeba castellanii* cysts through the excystation method. The conclusions highlight that the addition of FBS and 5%CO₂ accelerate this process and could be used to determine whether cysts are viable after exposure to anti-amoebic compounds. The study only describes one species and strain of *Acanthamoeba* and it would be useful to compare different species and strains as it is known that different species have different excystation periods. In addition, the study would benefit from a cyst disinfection/anti-amoebic assay to determine if the assay can be used to identify live/dead cysts. In addition, there are a couple of technical queries:

Response: Thank you very much for your helpful and insightful comments. In this study, standard AC strains (ATCC 50492) and two clinical AC strains were used. These strains did not exhibit the same excystation speed. Future studies will expand to include species beyond *Acanthamoeba castellanii*.

Following your comments, as an additional analysis, we evaluated the accuracy of the LACC assay using a new rapid protocol for AC cysts after chemical treatment. A new table has been created accordingly, and additional information has been added to the Materials and Methods, Results, and Discussion sections. Although chemical treatment (i.e., exposure to benzalkonium chloride) reduced LACC, there was no difference between the values obtained using the new rapid protocol and the 7-day protocol. Thus, the new rapid protocol provides high accuracy for the LACC assay in chemically treated samples.

1. the encystation method described specifies that the trophozoites are exposed to Neff encystation medium for 6-7 days until cysts are observed and then these were harvested. How do the authors know that the cysts were mature? Typically, the cysts are washed in 0.5% SDS to verify maturity. This is not documented in the methodology section.

Response: We sincerely apologize for the lack of explanation. Any remaining *Acanthamoeba castellanii* trophozoites were solubilized by treatment with 0.5% sodium dodecyl sulfate (SDS) at room temperature for 10 min. This explanation has been added to the Materials and Methods section (lines 232-234), along with the following references: Aqeel Y, Siddiqui R, Iftikhar H, Khan NA. 2013. The effect of different environmental conditions on the encystation of *Acanthamoeba castellanii* belonging to the T4 genotype. *Exp Parasitol* 135:30-5; Baig AM, Iqbal J, Khan NA. 2013. In vitro efficacies of clinically available drugs against growth and viability of an *Acanthamoeba castellanii* keratitis isolate belonging to the T4 genotype. *Antimicrob Agents Chemother* 57:3561-7; Dudley R, Jarroll EL, Khan NA. 2009. Carbohydrate analysis of

Acanthamoeba castellanii. Exp Parasitol 122:338-43; Yousuf FA, Siddiqui R, Khan NA. 2013. Acanthamoeba castellanii of the T4 genotype is a potential environmental host for Enterobacter aerogenes and Aeromonas hydrophila. Parasit Vectors 6:169.

2. From the microscopy images, it looks like growth of trophozoites is also accelerated in FBS and 5% CO₂. How do the authors account for this?

Response: We appreciate your helpful and insightful comments. As you have pointed out, the addition of FBS and a 5.0% CO₂ culture condition may accelerate trophozoite growth. In the observations on days 2 and 3 of culture shown in Figure 2, active trophozoite proliferation was observed under 5.0% CO₂, and the addition of 5% FBS to the PYG medium further accelerated the proliferation rate. This acceleration of proliferation, along with the promotion of excystation, led to the development of a rapid and accurate method for quantifying AC cysts.

The above explanations have been provided in the Results and Discussion sections (lines 114-119 and 181-183).

3. Typically, it is expected that ATCC strains are axenic, but some bacteria can be present. Did the authors perform any screening for bacterial presence prior to the experiments?

Response: Thank you very much for pointing this out. Prior to the start of this study, PYG medium containing cultured AC trophozoites was centrifuged at 5,000 g for 10 min, and the pellet was plated on Brain Heart Infusion agar to confirm the absence of bacteria.

This explanation has been added to the Materials and Methods section (lines 223-225).

4. line 43 page 8 what does plateau mean, upper limit of what?

Response: We sincerely apologize for the confusing description. By "plateau" we mean "reaching a peak and then remaining constant." Following the reviewer's comment, we have revised the sentence to avoid using the term "upper limit."

Other comments:

1. line 79 page 5: use the word 'protist' instead of 'protozoan'

Response: Thank you for your helpful advice. Following your suggestion, we have changed “protozoan” to “protist.”

2. line 85 page 5: instead of 'pathogenic' use 'an opportunistic pathogen'. It can also cause other diseases. Not just AK.

Response: Thank you for your helpful advice. Following your suggestion, the text has been revised as follows: “*Acanthamoeba castellanii* (AC) is an opportunistic pathogen and primarily causes refractory Acanthamoeba keratitis (AK).”

Re: Spectrum01170-25R1 (**A rapid and accurate protocol for quantifying living *Acanthamoeba castellanii* cysts**)

Dear Dr. Ryohei Hirose:

Your manuscript has been accepted, and I am forwarding it to the ASM production staff for publication. Your paper will first be checked to make sure all elements meet the technical requirements. ASM staff will contact you if anything needs to be revised before copyediting and production can begin. Otherwise, you will be notified when your proofs are ready to be viewed.

Sincerely,
Michael Ginger
Editor
Microbiology Spectrum